# Mixture of Complementary Agents for Robust LLM Ensemble

## Abstract

Multi-AI collaboration—such as ensembling or debating large language models (LLMs)—is a promising paradigm for aggregating information and boosting performance. A foundational step in these pipelines is to feed the responses of several *proposer* LLMs into a *summarizer* LLM, which synthesizes a better answer. However, choosing which proposers to include is non-trivial. Existing approaches primarily focus either on accuracy (picking the strongest models) or diversity (ensuring variety), and often overlook the interactions among proposers and with the summarizer. We reframe proposer selection as a combinatorial selection problem akin to feature selection, where the value of an LLM lies in its *complementarity* with others. However, directly applying standard feature-selection algorithms is impractical in the LLM setting due to prohibitive time complexity. Motivated by this limitation, we explore an extensive range of computationally feasible, greedy-style selection algorithms that assess complementarity using a small labeled set. Our experiments validate complementarity as a guiding principle for proposer selection and identify methods that achieve the best performance–cost trade-offs in practice.

## 1 Introduction

As today's Large language model (LLM) ecosystem fragments into numerous models with diverse expertise, collaboration among LLMs has become promising and sometimes necessary for tackling emerging tasks such as mathematical reasoning (Du et al., 2023), code generation (Mahmud et al., 2025), and complex decision-making (Wu et al., 2023). A convenient instantiation is *ensemble after inference*, which aggregates the LLM outputs after the generation of full responses. This includes well-studied frameworks such as *LLM debating* (Du et al., 2023; Estornell & Liu, 2024; Chan et al., 2023), in which multiple models iteratively exchange arguments before a final decision is reached, and *mixture-of-agents (MoA)* (Wang et al., 2024a; Li et al., 2025), which uses layered and summarization schemes to combine diverse model outputs.

A fundamental step in the ensemble framework is inputting the responses of $N$ LLM-prompt pairs —the *proposers*[1]—into an aggregating LLM—the *summarizer*—which synthesizes a potentially improved answer. Selecting which proposers to include is therefore critical: for a large proposer pool, it is impractical and inefficient to input responses from every available proposer due to context-window limits and the degraded inference ability (Liu et al., 2023). Existing methods often choose a small set of proposers based on their independent performance, primarily following two heuristics: (i) *accuracy-seeking*—prioritize high-accuracy proposers or even a single top model with multiple samples (Li et al., 2025; Jiang et al., 2023), and (ii) *diversity-seeking*—explicitly mix heterogeneous outputs or prompts to avoid reinforcing similar mistakes (Lau et al., 2024; Wang et al., 2024a).

However, both heuristics overlook a decisive factor: the complementarity of proposers both with one another and with the summarizer. We argue these team effects, rather than individual quality or pairwise diversity, ultimately determine ensemble performance. In particular, accuracy-seeking methods rank proposers only by their individual performance, while diversity-seeking methods reward variance regardless of quality. We instead propose *mixture-of-complementary-agents (**complementary-MoA**)*—a framework that selects proposers for how well they work together as a team and with the summarizer. The importance of complementarity

---

[1]The same LLM under different prompts can be viewed as different proposers.

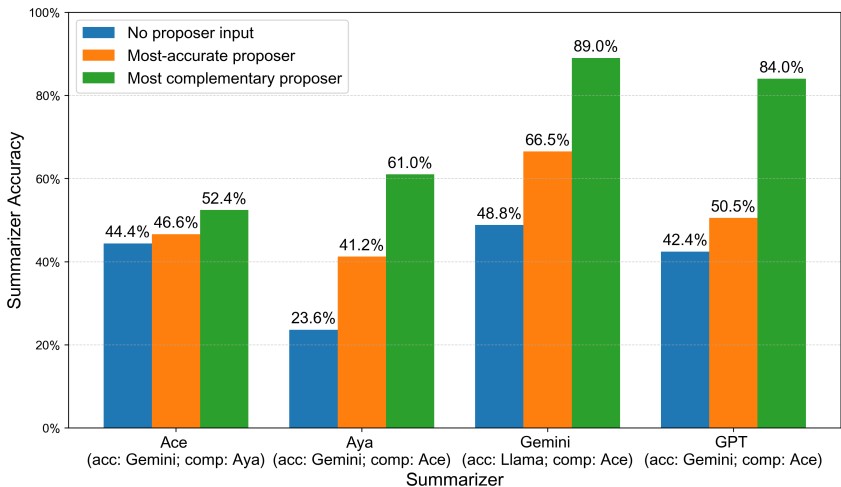

Figure 1: Summarizer accuracies on AIME (dolbokostya, 2025) when inputting the **most accurate** proposer vs. the **most complementary** proposer. For each summarizer $s$, the proposer pool is {Qwen3-32B, Sky-T1-32B-Preview, Aya-expanse-32B, Gemini-1.5-Pro, Llama-3.3-70B-Instruct, AceReason-Nemotron, and GPT-4o}, excluding $s$ itself.

can be observed from Figure 1, which compares summarizer accuracy when inputting (i) the individually most accurate proposer versus (ii) the proposer that most complements the summarizer. In this example, we consistently observe a nontrivial gap between the two choices, and furthermore, the most-complementary proposer is sometimes weak on its own. The upshot is both a promise and a challenge: as the ensemble size $k$ (the number of proposers selected for input) grows, the optimal selection can yield substantial gains, yet it also complicates the search, since optimal teams cannot be inferred from individual performance alone.

In this paper, we study **efficient proposer selection for multi-LLM ensembling** given a summarizer, with a focus on selecting complementary proposers rather than merely strong or diverse ones. Although proposer selection bears superficial resemblance to classical problems such as feature selection or data acquisition, the LLM setting introduces fundamentally new challenges: each evaluation of a candidate proposer team requires expensive summarizer calls, so wrapper-style methods are computationally prohibitive at scale. As a result, existing methods cannot be directly applied, and naive adaptations are prohibitively slow or brittle in practice. See Section 2 for more discussions.

Using multiple-choice QA as a concrete testbed, we frame proposer selection as a complementarity-driven optimization problem where the goal is to select a small ensemble that maximizes downstream accuracy while minimizing summarizer calls. This formulation captures both cross-model diversity and intra-model prompt variation, which prior work has shown to be a major source of performance gains (Li et al., 2025; Lau et al., 2024).

Motivated by this perspective, we develop a family of proposer-selection algorithms that navigate the accuracy–efficiency trade-off. We first introduce *model-first greedy*, a wrapper-style method that keeps the summarizer in the loop but reduces query complexity, measured by the total number of summarizer calls required for proposer selection. We select the winning model at each step based on the average marginal gain its prompt variants provide to the current ensemble $S$. Once a model is selected, we identify its best-performing prompt instance and add that model-prompt pair to $S$. Model-first greedy reduces query complexity based on the intuition that diversity across models matters more than diversity induced by different prompts within the same model.

To further reduce query complexity, we propose two algorithms that consider label-level complementarity. *Truth-prediction greedy* selects proposers based on how well their reported labels help predict the ground truth; *oracle-surrogate greedy* first fits a simple surrogate of the oracle and then selects proposers based

on their marginal contributions measured by the surrogate model. Both methods rely only on label-level statistics and therefore require no—or only light—summarizer calls.

We conduct an extensive empirical study across three popular reasoning benchmarks, spanning multiple proposer pools (a dominating-LLM regime and a mixed-crowd regime), different summarizers, and a range of ensemble sizes. This evaluation reveals a consistent pattern: commonly used heuristics perform well only in narrow regimes and fail unpredictably elsewhere. In contrast, our complementarity-guided methods, including the truth-prediction greedy which requires no summarizer call, are consistently robust across all scenarios. Moreover, we frequently observe substantial gains from model-first greedy over the strongest baseline, underscoring that explicitly optimizing for complementarity is crucial in ensemble frameworks.

In summary, our main contributions are threefold:

- We identify complementarity as a key, yet overlooked objective in agent-level LLM ensembles and propose a more principled proposer-selection framework, called complementary-MoA, that explicitly optimizes it.

- Inspired by feature-selection methods, we present three complementarity-driven selection algorithms designed around LLM ensembles, where evaluating each proposer requires expensive model calls. The resulting methods span a spectrum of accuracy–efficiency trade-offs, giving practitioners a principled way to choose under different query budgets.

- Through large-scale experiments, we provide a systematic and comprehensive comparison of LLM-compatible proposer-selection strategies, revealing the failure modes of existing heuristics and demonstrating that complementarity-based selection delivers the most reliable performance across all tested settings.

## 2 Related Work

**Agent-Level Ensemble.** LLM ensembles can be constructed at multiple stages of the inference pipeline (Chen et al., 2025b). We focus on *agent-level* ensembling, which treats each LLM as a black box. A closely related paradigm is *mixture-of-agents (MoA)* (Wang et al., 2024a), a layered collaboration scheme in which, at a given layer, multiple proposers submit responses that are then aggregated by a summarizer. Wang et al. (2024a) show that MoA effectively aggregates complementary signals, often yielding more reliable outputs than a single stronger model. A follow-up study challenges this design by demonstrating that repeatedly querying a single powerful LLM can also boost MoA-style performance (Li et al., 2025).

Another line of related work is *LLM debate* (Du et al., 2023; Estornell & Liu, 2024; Chan et al., 2023; Wang et al., 2023a; Baek et al., 2026), where multiple models iteratively critique and refine one another's responses that can often result in a consensus outperforming a single model. However, as Estornell & Liu (2024) point out, sharing all agents' responses is not always optimal, where they observe that selecting a subset of LLMs that maximizes the mutual information between agents can be more effective. Our work targets the foundational step in the above frameworks—the $N \rightarrow 1$ summarization—with particular emphasis on proposing a more principled way to decide which proposers to select for the summarizer.

Conditional mutual information is also a powerful tool for evaluating the informativeness of an agent's marginal contribution beyond the current information (Lu et al., 2024; Zhang et al., 2025). In concurrent work that applies a related idea to ensemble selection, Turkmen et al. (2026) explore complementarity in LLM ensembles by greedily maximizing the marginal mutual information between the proposer's suggested labels and the ground truth. While our high-level objectives align, their method is restricted to binary labels and does not account for the synergy between proposers and the summarizer. As illustrated in Sections 3.2 and 5.2, the summarizer significantly influences the composition of the optimal proposer set — a factor that label-level selection algorithms cannot capture.

**Training-Based Ensemble.** Prior literature has also explored the idea of training parametric meta-models to decide, per query, which LLM (or which LLM's output) to trust. For example, fusion methods train a small network on features from multiple LLMs—e.g., concatenated probabilities or last-layer embeddings—to

predict the true label (Jiang et al., 2023; Wang et al., 2023b). Routing methods learn a delegator that selects the most suitable agents for various tasks, e.g., RouteLLM uses human preference data to better trade off cost and quality (Ong et al., 2024), and ZOOTER learns a router based on distilling rewards on training queries Lu et al. (2023). Similarly, cost-aware cascades like FrugalGPT focus on learning when to use stronger but more expensive models (Chen et al., 2023). Unlike prior training-based ensembles, our framework avoids substantial supervised datasets: a few hundred examples suffice to learn the summarizer's behavior for better proposer selection. This light training also makes it compatible with closed-source LLM summarizers, whereas past work either does not use an LLM summarizer or requires open-source access (e.g., logits/weights).

**Feature-Selection.** Our problem is naturally relevant to feature selection, where the goal is to select a small subset of features that optimize the performance of an ML model. One of the most classic example is Wrapper (Kohavi & John, 1998), which evaluates features by repeatedly training a model—using forward/backward search. The selection of features can also be implemented by inducing sparsity during training, with examples like LASSO (Tibshirani, 1996) and LARS (Kolter & Ng, 2009). Furthermore, it is often beneficial to filter likely weak features without retraining a predictor based on information-theoretic (e.g., mRMR (Peng et al., 2005)) or neighborhood criteria (e.g., Relief (Urbanowicz et al., 2018)). However, two challenges limit applicability to our setting. First, wrapper-style methods demand extensive retraining, while filter/embedded approaches operate only at the label level and thus ignore the LLM summarizer's error-correcting behavior.[2] Second, the summarizer's performance is often non-monotone in the set of agents, making standard marginal-gain scoring unreliable; this motivates new evaluation metrics for agent contribution—e.g., our $k$-greedy algorithm in Section 4.2.

## 3    Problem Statement

We consider a dataset of multiple-choice questions $\mathcal{Q}$, and each question $q \in \mathcal{Q}$ has a ground-truth label $Y_q \in \mathcal{Y}$. We assume true labels are available on a validation subset $\mathcal{Q}_T \subset \mathcal{Q}$ of size $m = |\mathcal{Q}_T|$, while the remaining questions require inference (test data).

There are $N$ *proposers*. Proposer $i$ provides for question $q$ a response $R_{i,q} = (X_{i,q}, Z_{i,q})$, where $X_{i,q} \in \mathcal{Y}$ is a proposed label and $Z_{i,q}$ is textual supporting reasoning (e.g., chain-of-thought reasoning). In our setting, we permit multiple proposers to originate from a single LLM by varying the prompt. This is inspired by prior studies (Li et al., 2025; Lau et al., 2024), showing that feeding multiple responses from the same model to the summarizer can benefit the ensemble. Let $n_{\text{prompt}}$ and $n_{\text{LLM}}$ be the number of prompts and models; the total number of proposers is then $N = n_{\text{prompt}} \cdot n_{\text{LLM}}$.

To improve accuracy, a *summarizer* aggregates multiple proposer responses, and outputs a potentially more accurate label. Due to practical constraints (e.g., LLMs often have strict input context limits), we aim to select a (small) subset of proposers for the ensemble. Formally, given the ensemble size $k$ and a subset $S \subseteq [N]$ with $|S| = k$, the summarizer outputs $f(\boldsymbol{R}_{S,q})$, where $\boldsymbol{R}_{S,q} = (R_{i,q})_{i \in S}$. Both proposer and summarizer outputs are stochastic, and the key design choice is which proposers to select as input to the summarizer.

We evaluate a selection $S$ by the summarizer accuracy on test data:

$$\text{Acc}_f(S) = \frac{1}{|\mathcal{Q} \backslash \mathcal{Q}_T|} \sum_{q \in \mathcal{Q} \backslash \mathcal{Q}_T} \Pr\left[f(\boldsymbol{R}_{S,q}) = Y_q\right].$$

The central problem is to choose $k$ out of $N$ proposers to maximize accuracy, given summarizer $f$:

$$S^* = \arg\max_{S \subseteq [N], |S|=k} \text{Acc}_f(S). \tag{1}$$

We study the trade-offs introduced by the choice of $k$, though for clarity, most of our analysis proceeds while supposing $k$ is fixed.

---

[2]That said, in Section 5.2 we show that even label-level selection can produce more robust ensembles than existing baselines.

### 3.1 Previous Ideas

**Label-only aggregation.** The simplest approach aggregates only the discrete answers and ignores textual rationales. A common choice is (weighted) majority voting over all proposers or a selected subset. When proposers are conditionally independent with known accuracies, decision theory implies that a weighted majority rule (with weights proportional to log-odds of correctness) is optimal (Nitzan & Paroush, 1982). Our setting departs from these assumptions: LLM proposers exhibit strong dependencies, and their rationales carry additional signal. Empirically, aggregation schemes that leverage an LLM summarizer to use rationales outperform simple majority vote on labels alone (Lau et al., 2024; Tekin et al., 2024). Our experiments further confirm this point (see Appendix B.1).

**Accuracy-seeking aggregation.** A widely used heuristic in LLM ensembling is to select proposers by their estimated individual accuracy, where the intuition is that proposers with higher accuracy contribute more reliable evidence on new instances. For example, one idea, called the *self-MoA*, is to sample multiple diverse responses from the single best model and feed them to a summarizer (Li et al., 2025).

**Diversity-seeking aggregation.** A parallel line in the LLM ensemble literature argues that accuracy alone is insufficient: ensembles can benefit from diverse views. This intuition inspired several suggestions that explicitly encourage diversity or strike an accuracy–diversity trade-off (e.g., maximize diversity conditioned on an accuracy bar) (Lau et al., 2024; Tekin et al., 2024; Wang et al., 2024a). However, for LLM ensembles, such diversity-first strategies can be counterproductive: by admitting weak proposers in the name of variety, they often introduce low-quality or correlated errors that depress the final aggregation performance.

**LLM-as-a-Judge.** Another aggregation paradigm treats an LLM as a judge that evaluates, scores, or filters proposer responses and bases the final answer on the top-rated candidates or their summary. Such judge-based approaches can provide effective supervision without ground truth and have shown strong empirical performance (Liu et al., 2024). However, their effectiveness depends on the reliability and calibration of the judge model, and they may inherit systematic biases when judging responses from closely related LLMs.

### 3.2 Limitations of Previous Ideas: A Motivating Counterexample

Below we provide a counterexample with four proposers and one Bayesian summarizer, whose signals are $X_1, X_2, X_3, X_4$ and $Z_f$ respectively. We show that at $k = 2$, accuracy-seeking, mutual-information-seeking (Turkmen et al., 2026), and diversity-seeking algorithms all fail to identify the optimal $S^*$ in Equation (1) for ground truth $Y$.

**Proposition 1.** *There exists a joint distribution over $(Y, X_1, X_2, X_3, X_4, Z_f)$ and a Bayes-optimal summarizer $f$ such that $\mathrm{Acc}_f(\{1,2\}) = 1$ and $\mathrm{Acc}_f(S) \leq 0.9$ for every other $S \subseteq \{1,2,3,4\}$ with $|S| = 2$. However, none of the following methods select $\{1,2\}$:*

1. *Accuracy-first:* $\arg\max_{|S|=2} \sum_{i \in S} \Pr[X_i = Y]$.

2. *Mutual-information-seeking:* $\arg\max_{|S|=2} I(Y; X_S)$.

3. *Diversity-seeking:* $\arg\max_{i \neq j} \Pr[X_i \neq X_j]$.

The construction pits two types of proposer pairs against each other. Proposers $X_3$ and $X_4$ are individually informative about $Y$ (each agrees with $Y$ with probability 0.8) and conditionally independent given $Y$, making them look ideal under standard heuristics. Proposers $X_1$ and $X_2$, by contrast, are each marginally independent of $Y$ and highly correlated with one another, so they appear useless and redundant in isolation. The catch is that $Y = Z_f \oplus X_1 \oplus X_2$, so $X_1$ and $X_2$ become exactly informative once combined with the summarizer's private signal $Z_f$. Accuracy-first and mutual-information-seeking methods both prefer $\{3,4\}$ because $X_1, X_2$ have zero marginal accuracy and zero mutual information with $Y$; diversity-seeking also prefers $\{3,4\}$ because $X_1, X_2$ rarely disagree. Yet $\{1,2\}$ is the unique size-2 set that lets the Bayes-optimal summarizer recover $Y$ perfectly.

Although this construction is admittedly artificial, the message it conveys is clear and general: any selection rule that evaluates proposers without reference to $Z_f$ (the information of the summarizer) can miss the

complementarity that makes a selection optimal. Our empirical results in Figure 1 confirm this, motivating our framework of selecting proposers based on their complementarity with the summarizer beyond their individual quality or pairwise diversity.

## 4 Methods

Our central idea is to select proposers based on their collaborative performance with each other and with the summarizer—the selected proposers should complement their teammates. In principle, one could exhaustively evaluate all size-$k$ teams and pick the subset that maximizes summarizer accuracy. In practice, searching over all $\binom{N}{k}$ subsets is typically infeasible—e.g., even with $N = 20$ and $k = 5$ there are 15,504 candidate teams—especially given the high inference cost of summarizing multi-rationale inputs.

An immediate idea is a **greedy algorithm**: we can iteratively find the proposer with the largest marginal contribution to the summarizer accuracy until we find $k$ proposers. In particular, we initialize $S_0 = \emptyset$, and for $t = 1, \ldots, k$, choose

$$i_t \in \arg \max_{i \in [N] \setminus S_{t-1}} \left[ \mathrm{Acc}(S_{t-1} \cup \{i\}) - \mathrm{Acc}(S_{t-1}) \right],$$

then update $S_t = S_{t-1} \cup \{i_t\}$.

The performance of the greedy algorithm depends on the submodularity of the accuracy function, which in turn depends on the summarizer. It turns out that for LLM summarizers, the accuracy function is not even monotone (and thus not submodular)—including a low-accuracy proposer in the pool can actually reduce overall summarization performance. This observation is supported by prior work (Li et al., 2025) and our experiments in Appendix B.2. Therefore, in principle, the greedy algorithm can be far from the optimum in the worst case. However, as we will see, the empirical performance of the (simplified versions of) the greedy algorithm is generally robust and significantly outperforms the baselines.

Although the greedy algorithm is conceptually simple, it can be computationally demanding: it requires evaluating the accuracy function $O(kN)$ times, which entails $O(kNm)$ calls to the summarizer. It is thus important to explore the trade-off between ensemble accuracy and efficiency via some heuristic variants. In our experiments, we implement only the simplified methods rather than the full greedy algorithm.

### 4.1 Model-First Greedy

Recall that a model and an instruction prompt determine a proposer. However, responses generated by different prompts of the same model are typically more similar than responses generated by different models under the same prompt (see Appendix B.1). Inspired by hierarchical feature selection (Ristoski & Paulheim, 2014), we introduce a simplification called *model-first greedy*. Unlike standard greedy—which estimates the marginal gain of every proposer using all $m$ questions at each iteration —model-first greedy scores all $n_{\mathrm{prompt}}$ proposers from the same model using the common set of $m$ questions, then chooses a proposer within the best model. Concretely, in iteration $t$:

1. Partition $\mathcal{Q}_T$ into a training set $\mathcal{Q}_T^{tr}$ and a validation set $\mathcal{Q}_T^{val}$ for cross validation.

2. For each model $i \in [n_{\mathrm{LLM}}]$, randomly assign each question $j \in \mathcal{Q}_T^{tr}$ to one of the proposers associated with model $i$, and estimate the accuracy of model $i$ by averaging over the questions in the training set.

3. Select the model with the highest estimated accuracy, and within that model, pick the proposer with the highest estimated accuracy from the previous step.

Intuitively, the procedure prioritizes model selection while allowing more randomness in prompt selection. This reduces summarizer calls per iteration from $N \cdot m$ to $n_{\mathrm{LLM}} \cdot m$.

### 4.2 Label-level Complementarity

Model-first greedy estimates each proposer's marginal contribution via direct calls to the summarizer. However, the proposers' labels themselves carry predictive signals: the summarizer is more likely to answer correctly

when it receives more correct inputs. This motivates the idea of selecting proposers based on their *label-level* information, which can improve scalability by avoiding extensive calls to the summarizer oracle. This idea is related to *filter-based* feature selection methods, e.g. (Peng et al., 2005; Urbanowicz et al., 2018), which remove likely weak features without retraining the predictor based on correlations between features. In particular, we use an alternative set function $\widehat{Acc}$, defined with respect to a label-based summarizer $g$, and use it to guide proposer selection. This yields the following two methods.

**Truth-Prediction Greedy**  Built on the intuition that labels from a set of complementary proposers can predict the true label more accurately, we can train a light-weight machine learning model to predict $Y_q$, and use it to select informative proposers. Given a set of proposers $S$ and a family of models parametrized by $\theta \in \Theta$, we compute a value $\widehat{\mathrm{Acc}}_{g_\theta}(S)$ using the following procedure:

1. Partition $\mathcal{Q}_T$ into a training set $\mathcal{Q}_T^{tr}$ and a validation set $\mathcal{Q}_T^{val}$ for cross validation.

2. **Fit** $g_\theta$. Use the data in the training set, $((X_{i,q})_{i\in S}, Y_q)_{q \in \mathcal{Q}_T^{tr}}$ to fit a model $g_\theta$ that maps $|S|$ labels on a question to a (hard) prediction of the ground truth label. Here, proposers' generated labels are viewed as features.

3. **Score proposer set** $S$. On the validation set, evaluate the accuracy of $g_\theta$ using responses from $S$ and return $\widehat{\mathrm{Acc}}_{g_\theta}(S)$.

Next, we select proposers using a variant of the greedy algorithm, called the $k$-greedy (Alg. 1), using $\mathrm{Acc}_{g_\theta}$ as the set function. We first initialize a set of proposers $S_0 = \emptyset$. Then, in round $t \in \{1, \ldots, k\}$, unlike standard greedy—which estimates a candidate's marginal gain relative to the current selected set $S_{t-1}$—$k$-greedy's estimation always conditions on a set of $k$ proposers. The intuition is that LLM summarizers are non-monotone, so an element that looks promising early can hurt performance at the final team size $k$. Concretely, given $S_{t-1}$, we randomly select $k-t+1$ proposers, to form a team of size $k$, denoted as $L$. Then, we measure candidate $i$'s contribution ($i \notin S_{t-1}$) as the accuracy difference with and without $i$ (replacing one randomly chosen proposer in $L$). Averaging this difference over several random completions yields a more faithful estimate of $i$'s value at the final team size. We refer to this method as *truth-prediction greedy*, which applies the $k$-greedy algorithm to the set function $\widehat{\mathrm{Acc}}_{g_\theta}$. We emphasize that truth-prediction greedy relies on a lightweight ML model to guide proposer selection, but the final ensemble is still formed by feeding the chosen proposers into the LLM summarizer.

---

**Algorithm 1:** $k$-Greedy Proposer Selection w.r.t. Acc

**Input:** ground set $[N]$, target size $k$, set function Acc, repetitions $M$
**Output:** selected set $S_k$
$S_0 = \emptyset$ ;                                          // initial selected proposers
**for** $t = 1$ **to** $k$ **do**
    **for** $i \in [N] \setminus S_{t-1}$ **do**
        Set $\Delta_i = 0$;
        **for** $\tau = 1$ **to** $M$ **do**
            Sample $L \subseteq [N] \setminus (S_{t-1} \cup \{i\})$ uniformly with $|L| = k - |S_{t-1}|$;     // random completions
            Sample $j \in L$ uniformly and set $L' \leftarrow (L \setminus \{j\}) \cup \{i\}$;
            $\Delta_i \mathrel{+}= \mathrm{Acc}(S_{t-1} \cup L') - \mathrm{Acc}(S_{t-1} \cup L)$;
        $\widehat{\Delta}_i(S_{t-1}) = \Delta_i/M$ ;                                   // estimated marginal
    Choose $i^\star \in \arg\max_{i \in [N] \setminus S_{t-1}} \widehat{\Delta}_i(S_{t-1})$;
    $S_t = S_{t-1} \cup \{i^\star\}$;
**return** $S_k$;

---

**Oracle-Surrogate Greedy**  Proposer selection under truth-prediction greedy does not depend on the summarizer, so it may diverge from the ensemble's true test performance. As an alternative approach, we

propose *oracle-surrogate greedy*, where the idea is to fit a simple surrogate model to simulate the summarizer's behavior using a small number of oracle queries on the training set, then use the surrogate to score and select proposers. Although this method requires some summarizer calls for training, the surrogate model is kept simple as we only focus on label-level information, making it more sample-efficient than model-first greedy in practice.

We consider a surrogate model $\tilde{g}$ based on the assumption that the summarizer's accuracy depends primarily on how many of the $k$ input labels are correct. Specifically, $\tilde{g} : \{0, \ldots, k\} \to [0, 1]$ maps a count $c$ of correct labels to the expected summarizer accuracy when exactly $c$ out of $k$ inputs are correct. This implies that our surrogate model greatly reduces the query complexity by not distinguishing the proposer ID. Given a set of proposers $S$, the following procedure returns a value $\widehat{\text{Acc}}_{\tilde{g}}(S)$ for set $S$:

1. Partition $\mathcal{Q}_T$ into a training set $\mathcal{Q}_T^{tr}$ and a validation set $\mathcal{Q}_T^{val}$ for cross validation.

2. **Fit $\tilde{g}$.** For each $c \in \{0, \ldots, k\}$, repeat $T_{\tilde{g}}$ times: (i) sample a question from $\mathcal{Q}_T^{tr}$ and a size-$k$ set of proposers whose responses contain exactly $c$ correct labels; (ii) query the summarizer on these $k$ responses. Define $\tilde{g}(c)$ as the empirical accuracy—i.e., the average correctness of the summarizer across the $T_{\tilde{g}}$ queries.

3. **Score proposer set $S$.** For each $q \in \mathcal{Q}_T^{val}$, compute $c_q(S)$, the number of correct labels in $S$, and assign $\widehat{\text{Acc}}_{\tilde{g}}(S) = \frac{1}{|\mathcal{Q}_T^{val}|} \sum_{q \in \mathcal{Q}_T^{val}} \tilde{g}(c_q(S))$.

Next, we select a set of $k$ agents by calling Alg. 1 with $\widehat{\text{Acc}}_{\tilde{g}}(S)$ as the set function.

# 5 Experiments

In this section, we first introduce the experimental setups, then we validate the proposed complementary-MoA framework, diagnose the failure modes of baseline selectors, quantify efficiency–accuracy trade-offs, and finally study prompting strategies for the summarizer that yield stronger ensembles.

## 5.1 Experiment Setups

**Dataset** We consider benchmark datasets with multi-choice reasoning questions. We use three popular reasoning datasets: **AIME** (dolbokostya, 2025), CLadder (Jin et al., 2023), and MMLU-Pro (Wang et al., 2024b).

**AIME** consists of about 1,600 mathematical problems from competitions such as AIME and IMO. The original answers are open-ended integers; we convert each problem into a five-choice question by randomly sampling four incorrect integers between 0 and 1,000. **CLadder** contains 10k causal reasoning questions that translate causal-graph queries into natural-language yes/no questions, spanning association, intervention, and counterfactual reasoning. **MMLU-Pro** is a large-scale multi-choice benchmark targeting expert-level reasoning across diverse academic domains, with harder questions and reduced answer leakage compared to earlier MMLU versions.

**Models** We consider a diverse set of LLMs: QwQ-32B (Team, 2025b), Qwen3-32B (Team, 2025c), Sky-T1-32B-Preview (Team, 2025a), aya-expanse-32B (et al., 2024b), Gemini1.5-Pro (et al., 2024a), Llama-3.3-70B-Instruct (Dubey et al., 2024), AceReason-Nemotron (Chen et al., 2025a), and GPT-4o (et al., 2024c). All models are run with a default temperature of 0.7. To reduce inference cost and latency, we disable chain-of-thought prompting. For each of the LLMs, we consider $n_{\text{prompt}} = 5$ different prompts which are presented in Appendix C, and each model-prompt pair is viewed as a proposer.

**Settings** We evaluate ensemble performance across four factors—dataset, proposer pool, summarizer, and ensemble size $k$—using three datasets (CLadder, AIME, MMLU-Pro), two pools (with/without the strongest model), multiple summarizers, and several choices of $k \leq 5$. We use *complete pool* to refer to settings with all

tested LLMs, and *reduced pool* to refer to settings where the best-performing LLM is removed from the pool. We use the latter to simulate a setting with mixed-performing LLMs so as to improve the robustness of our results. For example, "(AIME, complete pool, Aya, $k = 3$)" denotes the AIME dataset, a pool including the strongest LLM, Aya as summarizer, and selecting three proposers. We limit $k$ to 5 because larger ensembles show diminishing returns (Lau et al., 2024), while the cost of searching for the optimal team grows quickly.

**Baselines**   Based on previous ideas in Section 3.1, we consider the following baselines:

- **Input-all**: input all $N$ proposers.[3]
- **Best-model**: identify the most accurate model and select all proposers associated with it, in line with (Li et al., 2025).
- **Top-accuracy**: select the most accurate $k$ proposers overall.
- **MoA (per-model top-1)**: for each model, select the single most accurate proposer, inspired by the original mixture-of-agents framework (Wang et al., 2024a).
- **Conditioned-diversity**: start with the most accurate proposer, then greedily add the proposer that maximizes average disagreement with the selected set, subject to an accuracy threshold $\tau = 0.4$, inspired by (Lau et al., 2024).
- **LLM-judge**: have a judge LLM (Aya or GPT-5.2) rate each proposer's responses on a 1–5 scale, and select the top-$k$ proposers by the average grade across the questions in training set. Cross validation is applied in the same manner as our methods descried in Section 4.
- **Approximate Shapley**: estimate each proposer's Shapley value, which is its average marginal contribution to summarizer accuracy across coalitions of other proposers. In particular, we sample 40 random subsets of sizes 1–4 (10 per size) and 20 random questions per subset, then averaging the accuracy gain from adding the proposer; select the top-$k$ proposers by the estimated Shapley value.

## 5.2   A Comparison of Proposer Selection Methods

We randomly select $m = 400$ questions for proposer selection and use the remaining questions for accuracy computing. For each LLM, we iteratively use $n_{\text{prompt}} = 5$ prompts to solicit responses for all the sampled questions, which returns $N = 40$ proposers' responses for each question. For each question, we randomize proposer order and include their individual accuracies in the instructions while inputting into the summarizer. To further reduce the variance of the ensemble accuracy (due to the randomness caused by the default temperature of LLMs), we repeatedly call the summarizer ten times for each question in the test set and take the average.

Table 1 presents a summary of accuracy results across six settings with $k = 5$. We tested two summarizers for each dataset while the better-performed one is presented in the figure: Aya for AIME, and AceReason for Cladder and MMLUPro. For the LLM-judge baseline, we report results under two judges of contrasting strength: Aya, a relatively weaker judge, and GPT-5.2, a strong judge. Detailed tables reporting the per-method composition of selected proposers for each setting are deferred to Appendix B.

**Importance of Complementarity**   First, our results show that accuracy-seeking and diversity-seeking baselines each fail in complementary ways. Accuracy-seeking methods (Top-accuracy, Best-model) perform poorly on AIME even when the pool contains a clearly strong proposer (Gemini1.5-pro). Consistent with Figure 1, this indicates that the most accurate proposer is not necessarily the most complementary to the summarizer. Diversity-seeking methods (Conditioned-diversity) instead fail on MMLU-Pro, where Sky alone composes well with the summarizer and adding diverse but lower-accuracy proposers degrades aggregation. In both cases, the failure stems from optimizing a proxy—individual accuracy or pairwise diversity—that does not capture how proposers interact with the summarizer.

In contrast, the label-level complementarity-aware methods (Truth-prediction Greedy and Oracle-surrogate Greedy), while not always the top performers, deliver consistently strong performance across all settings. Model-first Greedy goes further: it ranks among the top two methods in five of the six settings in Table 1.

---

[3]To fit the token limit, we truncate the responses from each proposer before summarizing.

Table 1: Accuracy comparison across all settings ($k = 5$). Per column, the best result is in **bold blue**, the second-best is in blue, and the worst is in red.

| Method | AIME (Aya) | | Cladder (AceReason) | | MMLU-Pro (AceReason) | |
|---|---|---|---|---|---|---|
| | Complete | Reduced | Complete | Reduced | Complete | Reduced |
| Input-all | **0.658** | 0.621 | 0.801 | 0.790 | 0.724 | 0.687 |
| Best-model | 0.349 | 0.332 | 0.786 | 0.793 | 0.722 | 0.664 |
| Top-accuracy | 0.377 | 0.364 | 0.777 | 0.780 | 0.746 | 0.666 |
| MoA | 0.580 | 0.562 | 0.757 | 0.760 | 0.737 | 0.687 |
| Conditioned-diversity | 0.483 | 0.468 | 0.742 | 0.765 | 0.683 | 0.669 |
| Aya-judge | 0.435 | 0.415 | 0.720 | 0.748 | 0.710 | 0.692 |
| GPT5.2-judge | 0.448 | 0.450 | 0.760 | 0.763 | 0.565 | 0.568 |
| Truth-prediction Greedy | 0.611 | 0.522 | 0.762 | 0.761 | 0.755 | 0.678 |
| Oracle-surrogate Greedy | 0.607 | 0.497 | 0.752 | 0.765 | **0.765** | 0.687 |
| Model-first Greedy | 0.654 | **0.632** | **0.812** | **0.802** | 0.738 | **0.711** |

Together, these findings indicate that explicitly accounting for complementarity is crucial for effective LLM ensembles.

The LLM-judge baseline performs poorly across all datasets. This is consistent with our main message: the judge evaluates each proposer in isolation rather than as a team interacting with the summarizer, and its grades can additionally inherit biases from the judge model itself. Notably, using a stronger judge does not necessarily yield a stronger ensemble—GPT5.2-judge is significantly outperformed by the weaker Aya-judge—reinforcing that individual-quality scoring is an unreliable proxy for ensemble value.

The approximate Shapley baseline is substantially more expensive than the other baselines, as it requires repeatedly sampling proposer subsets and isolating each proposer's marginal contribution. We therefore evaluate it only in the (AIME, ·, Aya, $k = 5$) setting for reference. As shown in Tables 12 and 13, its accuracy is consistently dominated by Model-first Greedy, indicating that Model-first Greedy offers a better trade-off between accuracy and query complexity.

*What explains this performance difference?* Figure 2 presents the empirical distributions of the number of correct labels $c \in \{0, \ldots, k\}$ obtained by the selected proposers under three representative methods on the AIME dataset with reduced pool. The bar corresponding to $c$ indicates the fraction of questions in the dataset where exact $c$ selected proposers have answered correctly. The overlaid curve shows the summarizer accuracy conditioned on $c$ out of $k$ input responses being correct.[4]

Clear patterns emerge: accuracy-seeking baselines, such as Best-model, induce a U-shaped distribution of $c$, while diversity-seeking baselines, such as Conditioned-diversity, exhibit a bell-shaped distribution. This indicates that Best-model tends to select proposers who make similar mistakes, which can be problematic when the summarizer accuracy curve is concave—i.e., when the marginal benefit of additional correct answers diminishes. However, Conditioned-diversity concentrates mass around $c = \lfloor k/2 \rfloor$ by seeking different proposers, which can be suboptimal when the summarizer requires a strong majority to achieve a significant accuracy boost. In contrast, complementarity-based methods yield distributions that lie between these two extremes, illustrating their robustness across different summarizer behaviors. This observation echoes our theoretical insights present in Section 3.2.

**Efficiency-Accuracy Tradeoff** We quantify each method's efficiency by the **number of summarizer calls** made during proposer selection (i.e., the query complexity). Calls used by individual proposers to generate responses are excluded, as they are identical across methods and do not affect relative efficiency. We summarize each method's complexity below, with formulas shown in Table 2.

---

[4]To reduce variance, we pool samples from all proposers to estimate the conditioned accuracy. Hence, the curve is identical across methods within the same setting.

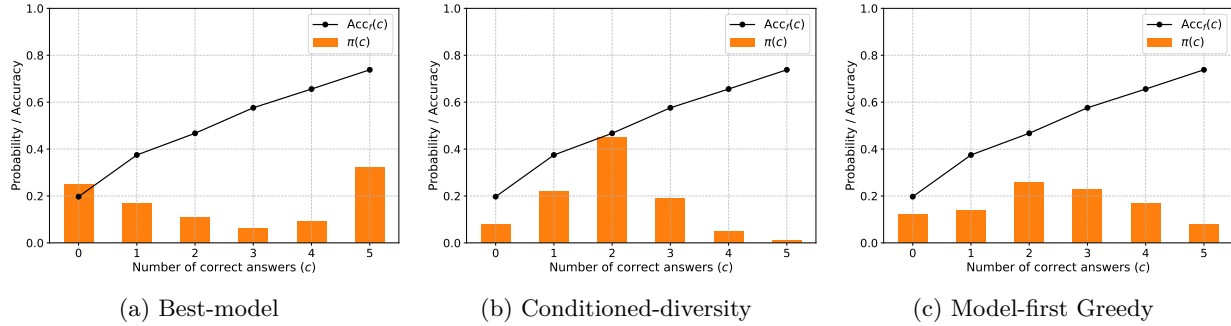

(a) Best-model        (b) Conditioned-diversity        (c) Model-first Greedy

Figure 2: Distribution of the number of correct answers (bars) and summarizer accuracy $\mathrm{Acc}_f(c)$ (line) for three exemplary methods in the (AIME, reduced pool, Aya, $k = 5$) setting.

Table 2: Query complexity of considered methods, measured as the number of summarizer queries during proposer-selection.

| Method | Complexity | | Symbol | Meaning |
|---|---|---|---|---|
| Approximate Shapley | $2\,n_{\mathrm{LLM}}\,n_{\mathrm{prompt}}\,z\,T_h$ | | $n_{\mathrm{LLM}}$ | number of candidate LLMs |
| Truth-Prediction Greedy | 0 | | $n_{\mathrm{prompt}}$ | number of prompts per LLM |
| Oracle-Surrogate Greedy | $(k+1)\,T_{\tilde{g}}$ | | $k$ | number of selected proposers |
| Model-First Greedy | $n_{\mathrm{LLM}}\,m\,k$ | | $m$ | size of training set |
| Other methods | 0 | | $z, T_h, T_{\tilde{g}}$ | method-specific Monte Carlo sample size |

- **LLM-judge baselines.** These methods issue no summarizer calls but require $m$ calls to a judge LLM to score proposer responses on the training set. We treat judge calls as substantially cheaper, since each produces only short numerical ratings, whereas a summarizer call must reason over a long, multi-rationale context and emit a full answer.

- **Oracle-Surrogate Greedy.** Fits the summarizer's accuracy by drawing $T_{\tilde{g}}$ Monte Carlo samples for each of the $(k+1)$ possible correct-count cases (from 0 to $k$), resulting in $(k+1)\,T_{\tilde{g}}$ calls; with $T_{\tilde{g}} = 200$ and $k = 5$, this is 1,200 calls.

- **Model-First Greedy.** Evaluates each of the $n_{\mathrm{LLM}}$ models on $m$ training questions in each of $k$ rounds, giving $n_{\mathrm{LLM}} \cdot m \cdot k$ calls. In our experiments, this is $8 \cdot 400 \cdot 5 = 16{,}000$ calls.

- **Approximate Shapley.** Estimates each proposer's Shapley value by random coalition sampling. For each of the $N = n_{\mathrm{LLM}} \cdot n_{\mathrm{prompt}}$ proposers $i$, we sample $z = 10(k-1)$ coalitions of other proposers (10 per coalition size, for sizes 1 to $k-1$), and for each sampled coalition $S$ we query the summarizer on $T_h = 20$ training questions twice—once on $S$ and once on $S \cup \{i\}$. The total is $2 \cdot N \cdot z \cdot T_h$ calls; with $k = 5$, this is $2 \cdot 40 \cdot 40 \cdot 20 = 64{,}000$ calls.

- **Other methods.** Other baselines and Truth-Prediction Greedy rely on proposers' individual reported labels and thus incur zero LLM calls during proposer selection.

### 5.3 Prompting Summarizer

Beyond which proposers are selected, how their responses are presented to the summarizer also shapes aggregation quality. We examine two prompting choices: (i) the ordering of proposer responses and (ii) whether each proposer's individual accuracy is disclosed to the summarizer.

We pick five proposers with relatively large accuracy differences, and input their responses to the summarizer in ascending, descending, or randomized order of individual accuracy. For each case, we further distinguish two settings depending on whether the accuracy of each proposer is input to the summarizer as a part of the prompt.

Table 3 presents an example with two key takeaways. First, **inputting accuracy matters.** Providing per-proposer accuracies affects performance in opposite ways across datasets—improving on AIME and MMLU-Pro yet degrading on CLadder. This suggests that the LLM summarizer can respond to the "reliability" information, but the net effect is heavily context-dependent. When accuracy information helps, the gain is largest under random ordering — consistent with the summarizer relying more on explicit reliability cues when no positional signal is available.

Second, **ordering matters.** Placing stronger proposers later in the prompt (using ascending accuracy order) outperforms descending order. This pattern is consistent with recency bias in long-context LLM inference (Peysakhovich & Lerer, 2023): earlier content tends to receive less attention relative to later content. These findings help clarify why our main experiments adopted a randomized ordering with per-proposer accuracies.

Table 3: Summarizer accuracies under different proposer orderings and whether individual accuracies are input in the $(\cdot, \cdot, \text{AceReason}, k = 5)$ setting with proposers: QwQ, Gemini, Llama, GPT, Aya, under instruction prompt 1 (Appendix C).

|  | AIME | | CLadder | | MMLU-Pro | |
| --- | --- | --- | --- | --- | --- | --- |
| Ordering | no acc | with acc | no acc | with acc | no acc | with acc |
| Ascending | 0.524 | 0.526 | 0.806 | 0.773 | 0.406 | 0.449 |
| Descending | 0.500 | 0.496 | 0.792 | 0.759 | 0.385 | 0.449 |
| Randomized | 0.498 | 0.538 | 0.798 | 0.774 | 0.401 | 0.448 |

## 6 Conclusion and Discussion

We propose **complementary-MoA**, a proposer-selection framework for post-inference LLM ensembles built around an observation that prior work has largely missed: an optimal ensemble depends not only on proposers' individual accuracy or mutual diversity, but on how well they complement both each other and the summarizer. Existing heuristics—whether based on individual accuracy or pairwise diversity—evaluate proposers in isolation from the summarizer and therefore overlook this structural factor in aggregation. We make the gap precise both theoretically and empirically: Proposition 1 constructs an example in which accuracy-, mutual-information-, and diversity-seeking selectors all simultaneously fail to recover the optimal proposer set, under a Bayes-optimal summarizer, and our experiments confirm that this is not a contrived artifact but a recurring failure mode in practice.

Building on this insight, we connect proposer selection to feature selection over a black-box objective and instantiate three algorithms—model-first, truth-prediction, and oracle-surrogate greedy—that span a spectrum of accuracy–efficiency trade-offs. Across three reasoning benchmarks, multiple summarizers, and varying proposer pools, our methods are consistently robust, clarify when and why standard baselines fail, and motivate prompting strategies that further strengthen multi-LLM collaboration.

We acknowledge several limitations and directions for future work. First, our label-level algorithms target multiple-choice tasks; extending them to open-ended generation requires a reliable evaluation metric that captures partial correctness. Second, the efficiency–accuracy frontier is not yet fully charted, where hybrid designs (e.g., selecting the first $k' < k$ proposers via complementarity and filling the remainder by accuracy) may further reduce query complexity.

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

## A    Proofs

*Proof of Proposition 1.* We consider binary signals in $\{0,1\}$. Let $X_1$ and $Z_f$ be independent and uniform on $\{0,1\}$, and set $X_2$ so that $\Pr[X_2 = X_1] = 0.9$. Define the ground truth as $Y = Z_f \oplus X_1 \oplus X_2$. Let $X_3, X_4$ be conditionally independent given $Y$ with $\Pr[X_3 = Y] = \Pr[X_4 = Y] = 0.8$. The summarizer privately observes $Z_f$, and proposer $i$ reports $X_i$ for $i = 1, \ldots, 4$.

**Optimality of $\{1, 2\}$.**    Because $Y$ is a deterministic function of $(Z_f, X_1, X_2)$, the Bayes-optimal summarizer recovers $Y$ exactly from $(Z_f, X_1, X_2)$, so $\mathrm{Acc}_f(\{1, 2\}) = 1$. Every other size-2 set contains at most one of $\{1, 2\}$ and hence cannot recover $Y$ exactly.

For $S = \{3, 4\}$, since $Z_f$ alone is independent of $Y$, it provides no information without $X_1$ or $X_2$. The Bayes-optimal $f$ therefore aggregates $X_3, X_4$ alone, agreeing on the correct value with probability 0.64 and tie-breaking on disagreement, for total accuracy

$$0.64 \cdot 1 + 0.32 \cdot \tfrac{1}{2} = 0.8.$$

For $|S \cap \{1, 2\}| = |S \cap \{3, 4\}| = 1$, without loss of generality, take $S = \{1, 3\}$. By symmetry it suffices to compute the conditional accuracy given $Z_f = X_1 = 0$, and the other three combinations are identical. Under this conditioning, $Y = X_2$, so $\Pr[Y = 0 \mid Z_f = 0, X_1 = 0] = 0.9$. The summarizer also observes $X_3$ with $\Pr[X_3 = Y \mid Y] = 0.8$. By Bayes' rule,

$$\Pr[X_3 = 0 \mid Z_f = 0, X_1 = 0] = 0.9 \cdot 0.8 + 0.1 \cdot 0.2 = 0.74,$$

and the posteriors on $Y$ are

$$\Pr[Y = 0 \mid Z_f = 0, X_1 = 0, X_3 = 0] = \frac{0.9 \cdot 0.8}{0.74} = \frac{36}{37},$$
$$\Pr[Y = 0 \mid Z_f = 0, X_1 = 0, X_3 = 1] = \frac{0.9 \cdot 0.2}{0.26} = \frac{9}{13}.$$

The Bayes-optimal outputs 0 regardless of $X_3$, and the conditional accuracy given $(Z_f = 0, X_1 = 0)$ is $\Pr[Y = 0 \mid Z_f = 0, X_1 = 0] = 0.9$. Hence $\mathrm{Acc}_f(\{1, 3\}) = 0.9$, and by symmetry the same holds for $\{1, 4\}, \{2, 3\}, \{2, 4\}$.

**Failure of the three methods.**    Since $Y$ is uniform conditional on $(X_1, X_2)$, we have $\Pr[X_1 = Y] = \Pr[X_2 = Y] = 0.5$ and $I(Y; X_1, X_2) = 0$. By contrast, $\Pr[X_3 = Y] = \Pr[X_4 = Y] = 0.8$ and a direct calculation gives

$$I(Y; X_3, X_4) = H(Y) - H(Y \mid X_3, X_4)$$
$$= 1 - \left(0.68 \cdot H\left(\tfrac{64}{68}\right) + 0.32 \cdot H\left(\tfrac{1}{2}\right)\right) > 0,$$

where $H(\cdot)$ denotes binary entropy and the two terms correspond to the agreement and disagreement events for $(X_3, X_4)$. Hence both accuracy-first and mutual-information greedy select $\{3, 4\}$.

For diversity-seeking, the pair $\{1, 2\}$ has disagreement $\Pr[X_1 \neq X_2] = 0.1$, which is strictly less than $\Pr[X_3 \neq X_4] = 0.32$. Hence the diversity-seeking never selects $\{1, 2\}$. □

## B  Additional Results

### B.1  Label-level Aggregation

In Table 4, 5, and 6, we present the accuracy of each proposer while answering the questions independently. As we can see, Aya is a weak model as a proposer, while we observe that it is a fast and accurate summarizer.

Table 4: Independent accuracy for each proposer with rows indicating models, columns indicating prompt IDs on the AIME dataset.

| Model | 1 | 2 | 3 | 4 | 5 |
|-------|-----|-----|-----|-----|-----|
| GPT-4o | 0.376 | 0.38 | 0.352 | 0.359 | 0.406 |
| AceReason-Nemotron-14B | 0.379 | 0.394 | 0.392 | 0.376 | 0.372 |
| Llama-3.3-70B-Instruct | 0.456 | 0.466 | 0.457 | 0.444 | 0.416 |
| QwQ-32B | 0.416 | 0.422 | 0.410 | 0.439 | 0.463 |
| Qwen3-32B | 0.408 | 0.439 | 0.403 | 0.42 | 0.437 |
| Sky-T1-32B-Preview | 0.387 | 0.398 | 0.381 | 0.383 | 0.388 |
| aya-expanse-32b | 0.233 | 0.248 | 0.253 | 0.235 | 0.267 |
| Gemini1.5-pro | 0.5 | 0.504 | 0.459 | 0.46 | 0.488 |

Table 5: Independent accuracy for each proposer with rows indicating models, columns indicating prompt IDs on the CLadder dataset.

| Model | 1 | 2 | 3 | 4 | 5 |
|-------|-----|-----|-----|-----|-----|
| GPT-4o | 0.681 | 0.680 | 0.682 | 0.677 | 0.685 |
| AceReason-Nemotron-14B | 0.708 | 0.726 | 0.692 | 0.707 | 0.726 |
| Llama-3.3-70B-Instruct | 0.507 | 0.595 | 0.502 | 0.538 | 0.532 |
| QwQ-32B | 0.775 | 0.789 | 0.803 | 0.802 | 0.797 |
| Qwen3-32B | 0.678 | 0.717 | 0.710 | 0.707 | 0.742 |
| Sky-T1-32B-Preview | 0.599 | 0.587 | 0.591 | 0.583 | 0.575 |
| aya-expanse-32b | 0.526 | 0.548 | 0.527 | 0.511 | 0.519 |
| Gemini1.5-pro | 0.707 | 0.704 | 0.718 | 0.734 | 0.748 |

We further test label-level aggregators against LLM summarizer aggregation, aiming to show that leveraging proposers' textual reasoning can boost accuracy. We evaluate three majority-vote variants: (i) over all proposers, (ii) over the best prompt per model, and (iii) over the best model per prompt. We also include weighted majority vote, using the classic log-odds weights $w_i \propto \log \frac{p_i}{1-p_i}$ derived for independent binary voters by Nitzan & Paroush (1982). Although our setting involves multiclass labels and correlated voters, we adopt this weighting as a heuristic baseline. Finally, we consider a learning-based baseline that trains a decision tree on the $N$ proposers' labels (features) to predict the ground truth, and report test accuracy.

As shown in Tables 7 to 9, majority-vote baselines perform competitively with the LLM summarizers on the binary CLadder dataset, yet they are outperformed by LLM summarizers on the multichoice AIME and MMLU-Pro datasets with the best method. These results underscore the importance of incorporating textual evidence from proposers' reasoning, rather than relying solely on label-level aggregation.

Table 6: Independent accuracy for each proposer with rows indicating models, columns indicating prompt IDs on the MMLU-Pro dataset.

| Model | 1 | 2 | 3 | 4 | 5 |
|---|---|---|---|---|---|
| GPT-4o | 0.45 | 0.416 | 0.444 | 0.45 | 0.418 |
| AceReason-Nemotron-14B | 0.519 | 0.525 | 0.521 | 0.539 | 0.507 |
| Llama-3.3-70B-Instruct | 0.473 | 0.463 | 0.444 | 0.454 | 0.478 |
| QwQ-32B | 0.582 | 0.589 | 0.573 | 0.576 | 0.577 |
| Qwen3-32B | 0.64 | 0.637 | 0.622 | 0.616 | 0.648 |
| Sky-T1-32B-Preview | 0.651 | 0.646 | 0.644 | 0.642 | 0.657 |
| aya-expanse-32b | 0.343 | 0.356 | 0.344 | 0.253 | 0.368 |
| Gemini1.5-pro | 0.549 | 0.53 | 0.536 | 0.537 | 0.533 |

Table 7: Label-level aggregation baselines on AIME.

| Method | Accuracy | |
|---|---|---|
| | Unweighted | Weighted |
| Majority | 0.627 | 0.644 |
| Majority (best prompt per model) | 0.583 | 0.597 |
| Majority (best model per prompt) | 0.580 | 0.580 |
| Decision Tree | 0.488 | |

## B.2 Comparing Proposer Selection Methods (Continued)

Here, we present the comparison of methods in other settings, aiming to show the robustness of our methods. We further present the composition of the selected proposer pool by each method to better illustrate their pros and cons.

**Ensemble Size** Table 11 and 10 report results for the (AIME, complete pool, Aya, ·) setting at $k \in \{3, 4\}$ settings. Note that the results for Input-all, Best-model, and MoA remain the same, as their performance does not depend on $k$. Our results confirm the robustness of our complementary-MoA framework, as it remains competitive with the strongest baselines. Overall, we do not observe a monotonic improvement in summarizer accuracy as the ensemble size increases.

**Other Settings** Here, we present the results for the remaining settings. Due to computational cost, we omit the LLM-judge and Approximate Shapley baselines in some settings; where they were evaluated, neither outperformed our methods. Two patterns emerge consistently across the tables. First, our complementarity-driven methods—particularly Model-first Greedy—rank first or second in nearly every setting, regardless of dataset, summarizer, pool composition, or ensemble size $k$. Second, the relative ordering of baselines is highly setting-dependent: Best-model is the strongest baseline for MMLU-Pro with Aya yet among the worst on AIME, and Input-all dominates AIME under Aya but underperforms under AceReason. This sensitivity is precisely the failure mode our framework targets, and the consistent strength of complementarity-aware selection across the same settings is the central empirical takeaway.

Table 8: Label-level aggregation baselines on CLadder.

| Method | Accuracy | |
|---|---|---|
| | Unweighted | Weighted |
| Majority | 0.795 | 0.814 |
| Majority (best prompt per model) | 0.816 | 0.816 |
| Majority (best model per prompt) | 0.793 | 0.805 |
| Decision Tree | 0.737 | |

Table 9: Label-level aggregation baselines on MMLU-Pro.

| Method | Accuracy | |
|---|---|---|
| | Unweighted | Weighted |
| Majority | 0.747 | 0.762 |
| Majority (best prompt per model) | 0.719 | 0.740 |
| Majority (best model per prompt) | 0.736 | 0.737 |
| Decision Tree | 0.439 | |

Table 10: A comparison of methods in the (AIME, complete pool, Aya, $k = 3$) setting. Per column, the best accuracy is in **bold blue**, the second-best is in blue, and the worst is in red.

| Method | Selected proposers (% of selections) | | | | | | | | Accuracy |
|---|---|---|---|---|---|---|---|---|---|
| | QwQ | Qwen | Llama | Gemini | GPT | Sky | Aya | AceReason | |
| Input-all | 12.5 | 12.5 | 12.5 | 12.5 | 12.5 | 12.5 | 12.5 | 12.5 | 0.645 |
| Best-model | 100 | – | – | – | – | – | – | – | 0.351 |
| Top-accuracy | 6.7 | – | 6.7 | 86.7 | – | – | – | – | 0.417 |
| MoA | 12.5 | 12.5 | 12.5 | 12.5 | 12.5 | 12.5 | 12.5 | 12.5 | 0.594 |
| Conditioned-diversity | – | – | 6.7 | 33.3 | – | – | 33.3 | 26.7 | 0.416 |
| Aya-dynamic | – | 35.3 | 52.9 | – | – | – | – | 11.8 | 0.414 |
| GPT5.2-dynamic | – | – | – | 40 | – | – | – | 60 | 0.470 |
| Truth-prediction Greedy | 13.3 | 33.3 | 13.3 | 33.3 | 6.7 | – | – | – | **0.714** |
| Oracle-surrogate Greedy | 40 | 6.7 | 20 | 33.3 | – | – | – | – | 0.528 |
| Model-first Greedy | 53.3 | 40 | – | 6.7 | – | – | – | – | 0.710 |

Table 11: A comparison of methods in the (AIME, complete pool, Aya, $k = 4$) setting. Per column, the best accuracy is in **bold blue**, the second-best is in blue, and the worst is in red.

| Method | Selected proposers (% of selections) | | | | | | | | Accuracy |
|---|---|---|---|---|---|---|---|---|---|
| | QwQ | Qwen | Llama | Gemini | GPT | Sky | Aya | AceReason | |
| Input-all | 12.5 | 12.5 | 12.5 | 12.5 | 12.5 | 12.5 | 12.5 | 12.5 | 0.746 |
| Best-model | 100 | – | – | – | – | – | – | – | 0.766 |
| Top-accuracy | 75 | 25 | – | – | – | – | – | – | 0.740 |
| MoA | 12.5 | 12.5 | 12.5 | 12.5 | 12.5 | 12.5 | 12.5 | 12.5 | 0.660 |
| Conditioned-diversity | 25 | – | 25 | – | 25 | 25 | – | – | 0.620 |
| Aya-dynamic | – | – | 35 | 60 | – | – | – | 5 | 0.424 |
| GPT5.2-dynamic | – | – | – | 87.1 | – | – | – | 12.9 | 0.472 |
| Truth-prediction Greedy | 50 | 50 | – | – | – | – | – | – | 0.740 |
| Oracle-surrogate Greedy | 50 | 25 | – | 25 | – | – | – | – | 0.728 |
| Model-first Greedy | 25 | 25 | – | – | – | – | – | 50 | **0.796** |

Table 12: A comparison of methods in the (AIME, complete pool, Aya, $k = 5$) setting. Per column, the best accuracy is in **bold blue**, the second-best is in blue, and the worst is in red.

| Method | Selected proposers (% of selections) | | | | | | | | Accuracy |
|---|---|---|---|---|---|---|---|---|---|
| | QwQ | Qwen | Llama | Gemini | GPT | Sky | Aya | AceReason | |
| Input-all | 12.5 | 12.5 | 12.5 | 12.5 | 12.5 | 12.5 | 12.5 | 12.5 | **0.658** |
| Best-model | – | – | – | 100 | – | – | – | – | 0.349 |
| Top-accuracy | 8 | – | 32 | 60 | – | – | – | – | 0.377 |
| MoA | 12.5 | 12.5 | 12.5 | 12.5 | 12.5 | 12.5 | 12.5 | 12.5 | 0.580 |
| Conditioned-diversity | – | – | 20 | 20 | – | – | 40 | 20 | 0.483 |
| Aya-dynamic | – | 32 | 60 | – | – | – | – | 8 | 0.435 |
| GPT5.2-dynamic | – | – | – | 64 | – | – | – | 36 | 0.448 |
| Approximate Shapley | – | – | – | – | – | – | – | 100 | 0.602 |
| Truth-prediction Greedy | 28 | 32 | 12 | 20 | – | – | – | 8 | 0.611 |
| Oracle-surrogate Greedy | 48 | 8 | – | 40 | – | – | – | 4 | 0.607 |
| Model-first Greedy | 32 | 32 | 4 | 4 | – | – | – | 28 | 0.654 |

Table 13: A comparison of methods in the (AIME, reduced pool, Aya, $k = 5$) setting. Per column, the best accuracy is in **bold blue**, the second-best is in blue, and the worst is in red.

| Method | Selected proposers (% of selections) | | | | | | | Accuracy |
| --- | --- | --- | --- | --- | --- | --- | --- | --- |
| | Qwen | Llama | Gemini | GPT | Sky | Aya | AceReason | |
| Input-all | 14.3 | 14.3 | 14.3 | 14.3 | 14.3 | 14.3 | 14.3 | 0.621 |
| Best-model | – | – | 100 | – | – | – | – | 0.332 |
| Top-accuracy | – | 40 | 60 | – | – | – | – | 0.364 |
| MoA (mixed) | 14.3 | 14.3 | 14.3 | 14.3 | 14.3 | 14.3 | 14.3 | 0.562 |
| Conditioned-diversity | – | 20 | 20 | – | – | 40 | 20 | 0.468 |
| Aya-dynamic | – | 32 | 60 | – | – | – | 8 | 0.415 |
| GPT5.2-dynamic | – | – | 59.3 | – | – | 33.3 | 7.4 | 0.450 |
| Approximate Shapley | – | – | – | – | – | – | 100 | 0.596 |
| Truth-prediction Greedy | 40 | 32 | 20 | 8 | – | – | – | 0.522 |
| Oracle-surrogate Greedy | 20 | 20 | 20 | – | 20 | – | 20 | 0.497 |
| Model-first Greedy | 32 | 8 | 8 | – | – | – | 52 | **0.632** |

Table 14: A comparison of methods in the (AIME, complete pool, AceReason, $k = 5$) setting. Per column, the best accuracy is in **bold blue**, the second-best is in blue, and the worst is in red.

| Method | Selected proposers (% of selections) | | | | | | | | Accuracy |
| --- | --- | --- | --- | --- | --- | --- | --- | --- | --- |
| | QwQ | Qwen | Llama | Gemini | GPT | Sky | Aya | AceReason | |
| Input-all | 12.5 | 12.5 | 12.5 | 12.5 | 12.5 | 12.5 | 12.5 | 12.5 | 0.313 |
| Best-model | – | – | – | 100 | – | – | – | – | 0.379 |
| Top-accuracy | 8 | – | 32 | 60 | – | – | – | – | 0.389 |
| MoA | 12.5 | 12.5 | 12.5 | 12.5 | 12.5 | 12.5 | 12.5 | 12.5 | 0.445 |
| Conditioned-diversity | – | – | 20 | 20 | – | – | 40 | 20 | 0.476 |
| Aya-dynamic | – | 32 | 60 | – | – | – | – | 8 | 0.436 |
| GPT5.2-dynamic | – | – | – | 64 | – | – | – | 36 | 0.487 |
| Truth-prediction Greedy | 12 | 36 | 20 | 20 | 12 | – | – | – | 0.465 |
| Oracle-surrogate Greedy | 36 | 8 | 16 | 36 | – | – | – | 4 | **0.502** |
| Model-first Greedy | 28 | 12 | 4 | 20 | – | – | 4 | 32 | 0.501 |

Table 15: A comparison of methods in the (AIME, reduced pool, AceReason, $k = 5$) setting. Per column, the best accuracy is in **bold blue**, the second-best is in blue, and the worst is in red.

| Method | Selected proposers (% of selections) | | | | | | | Accuracy |
| | Qwen | Llama | Gemini | GPT | Sky | Aya | AceReason | |
|---|---|---|---|---|---|---|---|---|
| Input-all | 14.3 | 14.3 | 14.3 | 14.3 | 14.3 | 14.3 | 14.3 | 0.344 |
| Best-model | – | – | 100 | – | – | – | – | 0.380 |
| Top-accuracy | – | 40 | 60 | – | – | – | – | 0.397 |
| MoA | 14.3 | 14.3 | 14.3 | 14.3 | 14.3 | 14.3 | 14.3 | 0.441 |
| Conditioned-diversity | – | 20 | 20 | – | – | 40 | 20 | 0.478 |
| Aya-dynamic | – | 32 | 60 | – | – | – | 8 | 0.439 |
| GPT5.2-dynamic | – | – | 59.3 | – | – | 33.3 | 7.4 | 0.472 |
| Truth-prediction Greedy | 44 | 24 | 28 | – | – | – | 4 | 0.466 |
| Oracle-surrogate Greedy | 16 | 8 | 76 | – | – | – | – | 0.406 |
| Model-first Greedy | 24 | 4 | 8 | 4 | 4 | 24 | 32 | **0.498** |

Table 16: A comparison of methods in the (Cladder, complete pool, AceReason, $k = 5$) setting. Per column, the best accuracy is in **bold blue**, the second-best is in blue, and the worst is in red.

| Method | Selected proposers (% of selections) | | | | | | | | Accuracy |
| | QwQ | Qwen | Llama | Gemini | GPT | Sky | Aya | AceReason | |
|---|---|---|---|---|---|---|---|---|---|
| Input-all | 12.5 | 12.5 | 12.5 | 12.5 | 12.5 | 12.5 | 12.5 | 12.5 | 0.801 |
| Best-model | 100 | – | – | – | – | – | – | – | 0.786 |
| Top-accuracy | 100 | – | – | – | – | – | – | – | 0.777 |
| MoA | 12.5 | 12.5 | 12.5 | 12.5 | 12.5 | 12.5 | 12.5 | 12.5 | 0.757 |
| Conditioned-diversity | 20 | – | 20 | – | – | – | 60 | – | 0.742 |
| Aya-dynamic | 52 | 48 | – | – | – | – | – | – | 0.720 |
| GPT5.2-dynamic | – | 40 | – | 60 | – | – | – | – | 0.760 |
| Truth-prediction Greedy | 40 | 20 | 8 | 20 | – | 4 | – | 8 | 0.762 |
| Oracle-surrogate Greedy | 60 | 12 | – | 28 | – | – | – | – | 0.752 |
| Model-first Greedy | – | – | – | 60 | 40 | – | – | – | **0.812** |

Table 17: A comparison of methods in the (Cladder, reduced pool, AceReason, $k = 5$) setting. Per column, the best accuracy is in **bold blue**, the second-best is in blue, and the worst is in red.

| Method | Selected proposers (% of selections) | | | | | | | Accuracy |
|---|---|---|---|---|---|---|---|---|
| | Qwen | Llama | Gemini | GPT | Sky | Aya | AceReason | |
| Input-all | 14.3 | 14.3 | 14.3 | 14.3 | 14.3 | 14.3 | 14.3 | 0.790 |
| Best-model | – | – | 80 | – | – | – | 20 | 0.793 |
| Top-accuracy | 20 | – | 40 | – | – | – | 40 | 0.780 |
| MoA | 14.3 | 14.3 | 14.3 | 14.3 | 14.3 | 14.3 | 14.3 | 0.760 |
| Conditioned-diversity | 8 | 20 | 12 | – | – | 60 | – | 0.765 |
| Aya-dynamic | 88 | – | – | – | 12 | – | – | 0.748 |
| GPT5.2-dynamic | 40 | – | 60 | – | – | – | – | 0.763 |
| Truth-prediction Greedy | 32 | 12 | 28 | – | – | – | 28 | 0.761 |
| Oracle-surrogate Greedy | 28 | – | 40 | 4 | – | – | 28 | 0.765 |
| Model-first Greedy | – | – | 56 | 44 | – | – | – | **0.802** |

Table 18: A comparison of methods in the (MMLU-Pro, complete pool, Aya, $k = 5$) setting. Per column, the best accuracy is in **bold blue**, the second-best is in blue, and the worst is in red.

| Method | Selected proposers (% of selections) | | | | | | | | Accuracy |
|---|---|---|---|---|---|---|---|---|---|
| | QwQ | Qwen | Llama | Gemini | GPT | Sky | Aya | AceReason | |
| Input-all | 12.5 | 12.5 | 12.5 | 12.5 | 12.5 | 12.5 | 12.5 | 12.5 | 0.377 |
| Best-model | – | – | – | – | – | 100 | – | – | **0.524** |
| Top-accuracy | – | 36 | – | – | – | 64 | – | – | 0.495 |
| MoA | 12.5 | 12.5 | 12.5 | 12.5 | 12.5 | 12.5 | 12.5 | 12.5 | 0.448 |
| Conditioned-diversity | – | 8 | 20 | – | 20 | 12 | 40 | – | 0.397 |
| Aya-dynamic | – | 60 | 36 | – | – | 4 | – | – | 0.462 |
| GPT5.2-dynamic | – | – | – | 100 | – | – | – | – | 0.455 |
| Truth-prediction Greedy | 8 | 40 | – | – | 4 | – | 48 | – | 0.485 |
| Oracle-surrogate Greedy | 4 | 56 | – | – | – | 40 | – | – | 0.487 |
| Model-first Greedy | 52 | 36 | 4 | – | 4 | – | – | 4 | 0.502 |

Table 19: A comparison of methods in the (MMLU-Pro, reduced pool, Aya, $k = 5$) setting. Per column, the best accuracy is in **bold blue**, the second-best is in blue, and the worst is in red.

| Method | Selected proposers (% of selections) | | | | | | | Accuracy |
| --- | --- | --- | --- | --- | --- | --- | --- | --- |
| | QwQ | Qwen | Llama | Gemini | GPT | Aya | AceReason | |
| Input-all | 14.3 | 14.3 | 14.3 | 14.3 | 14.3 | 14.3 | 14.3 | 0.346 |
| Best-model | – | 100 | – | – | – | – | – | 0.482 |
| Top-accuracy | – | 100 | – | – | – | – | – | 0.481 |
| MoA | 14.3 | 14.3 | 14.3 | 14.3 | 14.3 | 14.3 | 14.3 | 0.464 |
| Conditioned-diversity | – | 20 | 20 | – | 20 | 40 | – | 0.404 |
| Aya-dynamic | – | 64 | 36 | – | – | – | – | 0.460 |
| GPT5.2-dynamic | – | – | – | 100 | – | – | – | 0.455 |
| Truth-prediction Greedy | 4 | 80 | – | 12 | 4 | – | – | 0.478 |
| Oracle-surrogate Greedy | 12 | 76 | 4 | 8 | – | – | – | **0.487** |
| Model-first Greedy | 32 | 48 | 4 | 4 | – | – | 12 | 0.475 |

Table 20: A comparison of methods in the (MMLU-Pro, complete pool, AceReason, $k = 5$) setting. Per column, the best accuracy is in **bold blue**, the second-best is in blue, and the worst is in red.

| Method | Selected proposers (% of selections) | | | | | | | | Accuracy |
| --- | --- | --- | --- | --- | --- | --- | --- | --- | --- |
| | QwQ | Qwen | Llama | Gemini | GPT | Sky | Aya | AceReason | |
| Input-all | 12.5 | 12.5 | 12.5 | 12.5 | 12.5 | 12.5 | 12.5 | 12.5 | 0.724 |
| Best-model | – | – | – | – | – | 100 | – | – | 0.722 |
| Top-accuracy | – | 36 | – | – | – | 64 | – | – | 0.746 |
| MoA | 12.5 | 12.5 | 12.5 | 12.5 | 12.5 | 12.5 | 12.5 | 12.5 | 0.737 |
| Conditioned-diversity | – | 8 | 20 | – | 20 | 12 | 40 | – | 0.683 |
| Aya-dynamic | – | 60 | 36 | – | – | 4 | – | – | 0.710 |
| GPT5.2-dynamic | – | – | – | 100 | – | – | – | – | 0.565 |
| Truth-prediction Greedy | 4 | 52 | 4 | – | – | – | 40 | – | 0.755 |
| Oracle-surrogate Greedy | 20 | 40 | – | – | – | 40 | – | – | **0.765** |
| Model-first Greedy | 40 | 16 | – | 12 | 8 | 8 | 16 | – | 0.738 |

Table 21: A comparison of methods in the (MMLU-Pro, reduced pool, AceReason, $k = 5$) setting. Per column, the best accuracy is in **bold blue**, the second-best is in blue, and the worst is in red.

| Method | Selected proposers (% of selections) | | | | | | | Accuracy |
|---|---|---|---|---|---|---|---|---|
| | QwQ | Qwen | Llama | Gemini | GPT | Aya | AceReason | |
| Input-all | 14.3 | 14.3 | 14.3 | 14.3 | 14.3 | 14.3 | 14.3 | 0.687 |
| Best-model | – | 100 | – | – | – | – | – | 0.664 |
| Top-accuracy | – | 100 | – | – | – | – | – | 0.666 |
| MoA | 14.3 | 14.3 | 14.3 | 14.3 | 14.3 | 14.3 | 14.3 | 0.687 |
| Conditioned-diversity | – | 20 | 20 | – | 20 | 40 | – | 0.669 |
| Aya-dynamic | – | 64 | 36 | – | – | – | – | 0.692 |
| GPT5.2-dynamic | – | – | – | 100 | – | – | – | 0.568 |
| Truth-prediction Greedy | 4 | 80 | – | 12 | 4 | – | – | 0.678 |
| Oracle-surrogate Greedy | 20 | 64 | – | 12 | 4 | – | – | 0.687 |
| Model-first Greedy | 28 | 24 | – | 16 | 12 | 20 | – | **0.711** |

## C  Prompts

---

**Multi-choice — Proposer Prompt**

You will solve a multiple choice question. Format your answer to include:
1. A full response
2. A concise step-by-step reasoning
3. The single letter choice

---

**Binary-choice — Proposer Prompt**

You will answer a yes or no question. Format your answer to include:
1. A full response
2. A concise step-by-step reasoning
3. The yes or no answer

---

**Multi-choice — Summarizer Prompt**

I will give you a multiple choice question and potential solutions that may be correct or incorrect. Your task is to analyze the reasoning of the potential solutions step by step.
If there are any errors, correct them and update your answer.
If there are no errors, answer the question matching those solutions.
Your answer must be in the format of a full response, then a letter choice.

---

**Binary-choice — Summarizer Prompt**

I will give you a yes or no question and multiple potential solutions that may be correct or incorrect. Your task is to analyze the reasoning of the potential solutions step by step.
If there are any errors, correct them and update your answer.
If there are no errors, answer the question matching those solutions.
Your answer must be in the format of a full response, then a yes or no answer.

---

**Instruction Prompt 1**

Divide the question into smaller, manageable parts and tackle each part individually before synthesizing the overall answer.

---

**Instruction Prompt 2**

Use mathematical principles and logic to solve the problem, even if it's not a math question.

---

**Instruction Prompt 3**

Relate the question to a familiar concept or situation to better understand and solve it.

---

**Instruction Prompt 4**

Think about what the answer would be if the opposite were true, to gain a different perspective.

---

**Instruction Prompt 5**

Eliminate the obviously incorrect answers first and then choose the most likely correct answer.

## D    LLM Usage

Large language models (LLMs) were used in this paper only as a general-purpose writing assistant. Specifically, they supported adjusting phrasing for clarity, polishing grammar, shortening sentences, and reformatting text. LLMs were also used to generate and refine tables (e.g., aligning multi-column headers and converting between LaTeX table styles). At no point did LLMs contribute to research ideas, conceptual framing, or experimental design. All substantive intellectual contributions are solely those of the authors.

