# OpenReview forum: "Mixture of Complementary Agents for Robust LLM Ensemble"
_TMLR — Under review for TMLR_

### Review · Reviewer_SFe9 · 2026-07-03

**Summary Of Contributions:**

The paper studies which proposer responses to feed into a summarizer LLM in a post-inference ensemble. The authors argue that the standard heuristics, accuracy-seeking and diversity-seeking, both ignore a third factor: how well the selected proposers complement each other and the summarizer. They reframe proposer selection as a feature-selection style combinatorial problem and propose a framework they call complementary-MoA. The paper proposes three selection algorithms that span an accuracy vs query-cost spectrum: model-first greedy, truth-prediction greedy, and oracle-surrogate greedy. The authors take empirical comparisons across three reasoning benchmarks, two proposer pools with and without the strongest model, multiple summarizers, and ensemble different sizes of k, against eight baselines.

**Key strengths.** The complementarity framing is clean and the counterexample illustrates it well at the conceptual level. The experimental sweep is broad, the appendix reports per-setting proposer compositions, and the query-complexity analysis is concrete and practically useful.

**Key weaknesses.** The headline empirical gains are small and inconsistent, and there is no quantification of variability for differences that are often 1 to 2 points or less. The trivial Input-all baseline matches or beats the best proposed method in several settings. The proof of the only formal result (Proposition 1) does not go through as written, because the construction makes the summarizer's private signal correlated with the ground truth. The practical methods, especially the two label-level ones, do not actually capture the summarizer-conditioned complementarity that the theory is built around. And the main table presents only the "better-performed" summarizer per dataset, which understates how much summarizer choice drives the results.

**Audience:**

Yes

**Audience Explanation:**

Multi-LLM ensembling, mixture-of-agents, and LLM debate are active topics, and the specific question of which proposers to feed a summarizer under a query budget is practically relevant. The explicit query-complexity accounting and the finding that a label-only method with zero summarizer calls can be competitive in several settings are the kind of results practitioners would want to know. The summarizer-prompting observations in Section 5.3 are also of independent interest.

**Broader Impact Concerns:**

I do not see serious ethical concerns. The work is about aggregation accuracy on reasoning benchmarks and does not introduce new data collection on people or obvious dual-use risk. There is no Broader Impact Statement in the submission, which I think is acceptable for this kind of paper.

**Claims And Evidence:**

No

**Claims Explanation:**

Because the core empirical claims as stated are stronger than the evidence, are not accompanied by any variability analysis, the only formal result has an incorrect proof, and the theory-to-method bridge is overstated, I cannot mark the claims as currently supported. But I want to be clear that most of my concerns look fixable.

- "Substantial gains from model-first greedy over the strongest baseline" (Introduction, last paragraph before the contribution list) is contradicted by the paper's own Table 1. On AIME complete the trivial Input-all baseline (0.658) is higher than Model-first Greedy (0.654), so the best method does not beat the simplest no-selection baseline. On MMLU-Pro complete, Model-first Greedy (0.738) is only the fourth-best entry, below Oracle-surrogate (0.765), Truth-prediction (0.755), and even Top-accuracy (0.746). In the five settings where a proposed method is on top, the margin over the strongest baseline is about 1 to 2 points (0.011 on AIME reduced, 0.011 and 0.009 on the two CLadder columns, 0.019 on both MMLU-Pro columns). "Substantial" is not the right word for these margins.

- Proposition 1 is not correct as stated, and the proof in Appendix A relies on a false step. The proof asserts that "Zf alone is independent of Y". It is not: since Y = Zf xor X1 xor X2 and Pr[X2 = X1] = 0.9, the term X1 xor X2 equals 0 with probability 0.9, so Pr[Y = Zf] = 0.9. The summarizer therefore already reaches accuracy 0.9 with zero proposers. More importantly, for S = {3,4} the Bayes-optimal summarizer combines this 0.9 prior from Zf with the two 0.8-accurate signals X3 and X4 and reaches 0.928, not the 0.8 computed in the proof (it follows Zf unless both X3 and X4 contradict it; the correct mass is 0.576 + 0.288 + 0.064 = 0.928, and I verified this by exact enumeration of the joint distribution). This violates the proposition's claim that Acc_f(S) <= 0.9 for every other size-2 set. The mixed-pair value of 0.9 does check out and {1,2} is still the unique optimum, so the qualitative separation survives, but the theorem as written is false. There is a second, smaller problem in the Section 3.2 prose: it says diversity-seeking "also prefers {3,4}", but under the construction X1 and X3 are independent uniform bits, so every mixed pair has disagreement 0.5, larger than Pr[X3 != X4] = 0.32. The diversity criterion of Proposition 1 would therefore select a mixed pair, not {3,4}. The proof itself only shows {1,2} is never selected (which is right, Pr[X1 != X2] = 0.1 is the smallest), so the prose should be aligned with that.

- The theory and the practical methods are not the same object. Proposition 1 turns on identity-level, summarizer-conditioned structure: X1 and X2 each have 0.5 marginal accuracy and zero mutual information with Y, and only become useful jointly with Z_f. But the paper assumes the summarizer's accuracy depends only on the count c of correct labels, ignoring which proposers are correct. That method would not solve the Proposition 1 example at all, since X1 and X2 contribute to the count essentially at random. The same holds for truth-prediction greedy: with I(Y; X1, X2) = 0, no label-only model g_theta can score {1,2} above chance, so it would also prefer {3,4}. The paper effectively concedes this in Section 2, where it criticizes Turkmen et al. (2026) because label-level selection "cannot capture" the synergy with the summarizer, yet two of its own three methods are label-level. Only model-first greedy keeps the summarizer in the loop. Figure 2 even suggests the real mechanism is "shape the distribution of the number-correct to match the summarizer's accuracy curve," which is a coarser and different idea from the XOR complementarity in Section 3.2. The claim that the methods follow from the theoretical insight is therefore overstated.

- The main results selectively present one summarizer per dataset ("the better-performed one is presented", Section 5.2). The appendix shows summarizer choice changes everything: on AIME the same Input-all baseline is 0.658 with Aya (Table 12) but 0.313 with AceReason (Table 14). Aya is also a weak model (independent accuracy 0.23 to 0.27, Table 4), and feeding it five prompts of strong Gemini collapses to 0.349, far below Gemini's own 0.5. I cannot tell from the main table whether the method gaps survive under a strong summarizer or whether they depend on a weak summarizer that badly degrades homogeneous strong inputs. This needs to be shown, not deferred. For CLadder the situation is worse: I could not find the second summarizer's results anywhere, the appendix Tables 16 and 17 also cover only AceReason.

- No variability is reported. The summarizer is called ten times per test question and averaged (Section 5.2), and the selection algorithms are themselves stochastic (the composition tables, for example Table 12, show Model-first spreading its picks across five different models over runs). But there are no standard deviations, no confidence intervals, and no significance tests. With many cells differing by 0.004 to 0.02, the "ranks among the top two in five of six settings" claim could be partly within noise.

**Requested Changes:**

1. Report variability and significance. Add standard deviations or confidence intervals over the ten summarizer repeats and, separately, over the randomness of the selection procedure, for Table 1 and the appendix tables. State which differences are statistically meaningful.
2. Calibrate the gains language to the evidence. The phrase "substantial gains over the strongest baseline" should be reconciled with AIME complete (Input-all 0.658 >= Model-first 0.654) and MMLU-Pro complete (Model-first is fourth). Since Input-all uses all 40 proposers while the proposed methods use k=5, the real advantage is test-time inference cost, not accuracy. Please quantify the test-time savings (summarizer input tokens or calls at inference, and any accuracy retained) so the value proposition is stated in the dimension where it actually holds. Right now Section 5 measures only selection-time query complexity, not the inference-time cost that selecting k proposers is supposed to save.
3. Fix Proposition 1. The step "Zf alone is independent of Y" in Appendix A is false under the stated construction (Pr[Y = Zf] = 0.9), and consequently Acc({3,4}) = 0.928 rather than 0.8, which breaks the claimed bound of 0.9. Either restate the proposition or repair the construction. One possible repair is to weaken the correlation, e.g. Pr[X2 = X1] = 0.5 + epsilon for a small epsilon; my quick check with epsilon = 0.05 gives Acc({3,4}) = 0.816 and mixed pairs at 0.8, with all three heuristics still failing to select {1,2}, but the authors should verify. Also correct the Section 3.2 sentence saying diversity-seeking "prefers {3,4}": under the stated criterion it prefers a mixed pair (disagreement 0.5 vs 0.32), and the proof only establishes it never picks {1,2}.
4. Reconcile theory and methods. Explain explicitly which of the three proposed methods, if any, captures the summarizer-conditioned complementarity of Proposition 1, and acknowledge that both label-level methods would fail that example: oracle-surrogate because it only counts correct labels, and truth-prediction because I(Y; X1, X2) = 0 leaves nothing for a label-only model to learn. Either soften the claim that the methods operationalize the theoretical insight, or add a method or analysis that does.
5. Present both summarizers in the main results, or justify the selection. Move the second summarizer per dataset into Table 1 (or a companion table next to it) rather than only the appendix, and add a sentence on how summarizer strength affects the method ranking, since Tables 12 and 14 show it matters a lot. For CLadder, the second summarizer's results appear to be missing even from the appendix and should be added.
6. Discuss open-ended generation. The intro motivates with MoA, which targets open-ended tasks, but all experiments are multiple-choice or binary and the label-level methods depend on discrete labels. The conclusion acknowledges this in one sentence; a slightly longer discussion or a small pilot would clarify how far the findings transfer.
7. Specify the truth-prediction model g_theta (e.g., hyperparameters, how cross-validation is set up). The main text only says "a family of models parametrized by theta", which is not reproducible. Clarify how it differs from the decision-tree label baseline in Appendix B.1.

---

### Review · Reviewer_qoTh · 2026-07-10

**Summary Of Contributions:**

The paper asks: with a pool of proposer LLMs and room for only k of them, which do you pick? You could pick the most accurate, or most diverse set, but the authors argue that this misses how proposers complement each other and the summarizer. A weak proposer can still be the best teammate for a given summarizer.  They frame the problem as feature selection with a black-box objective with three greedy algorithms at different cost points: model-first greedy, truth-prediction greedy, and oracle-surrogate greedy.

**Strengths**
- The core point is valuable: proposer value depends on the summarizer, not just the proposer. Prop. 1 makes this crisp.
- Figure 2 is the most convincing part of the paper. The U-shaped vs. bell-shaped correct-count distributions actually explain why best-model and diversity-seeking fail in opposite regimes, rather than just showing that they do.
- The paper is honest that summarizer accuracy is non-monotone (so greedy has no guarantee), and I think estimating marginal value at the final team size makes sense.

**Weaknesses**
- The results tables contradict each other, which puts the empirical claims in doubt.
- No error bars, and many wins are too small to trust without them.
- The main method is underspecified and inconsistent with its own cost analysis

**Audience:**

Yes

**Audience Explanation:**

Anyone building MoA or multi-LLM pipelines runs into this exact question. You can only fit K proposers into the summarizer's context, so which ones do you pick? The paper's insights are useful even if you never use their algorithms: picking the most accurate proposers and picking the most diverse ones both fail, in opposite ways, and what actually matters is the fit with the summarizer. The fact that one of their methods needs zero summarizer calls also makes it easy to actually try.

**Broader Impact Concerns:**

None. no specific concerns beyond generic LLM considerations.

**Claims And Evidence:**

No

**Claims Explanation:**

- Some baselines don't depend on k. The paper says so itself. So their numbers should be identical across Tables 10–12, which only vary k. They aren't. Input-all goes 0.645 / 0.746 / 0.658, and Best-model jumps from 0.351 to 0.766 while picking a different model at k=5. Either the tables have errors, or the variance between runs is so large that most of the reported gaps could be noise. This needs to be fixed before the main comparisons can be trusted.
- The selection process is random and repeated (the "% of selections" columns imply multiple runs, but the count is never stated), yet every table reports a single number. Several key results differ by only 1–2 points (0.812 vs. 0.801 on CLadder). And weakness 1 suggests run-to-run variance is larg. So the claim of robustness isn't supported yet. Please report standard deviations over repeated runs.
- Table 12 shows five Gemini responses fed to Aya scoring 0.349. worse than Gemini alone (0.50). The summarizer destroying value is important and the authors don't discuss this.  So I'm confused about whether baseline failures are selection failures or summarizer related.

**Requested Changes:**

1. Fix or explain the table inconsistencies as mentioned in the above section.

2. Add standard deviations (or CIs)/error bars over repeated selection runs and summarizer sampling for all main tables with the number of runs. Revise/restructure claims that rest on 1–2 point margins, like CLadder (0.812 vs. 0.801) or AIME at $k=5$ where Input-all beats model-first greedy.

3. Rewrite Section 4.1 so model-first greedy matches its claimed query cost. The steps never invoke the summarizer, yet Table 2 charges 16,000 calls. Pseudocode would help. Also state the model class for $g_\theta$, the value of $M$ in Alg. 1, and $T_{\tilde{g}}$.

4. Justify not using chain-of-thought for all models. QwQ and Sky-T1 are reasoning models, so wouldn't chain-of-thought strengthen the claims?

5. Compare against Turkmen et al. on CLadder, which is binary. The paper dismisses their method as binary-only, but that's exactly where a comparison was feasible? If not, please explain why.

6. Discuss the Table 12 result, where five Gemini responses through Aya score 0.349, worse than Gemini alone. The summarizer destroying value is directly relevant to the paper's story.

7. Either expand Section 5.3 (more configurations, significance) or soften its claims.

8. Fix the LARS citation, the broken "et al." bibliography entries, and the "descried" typo in Section 5.1.

---

### Review · Reviewer_4bYr · 2026-07-13

**Summary Of Contributions:**

- The paper studies the problem of post inference LLM ensembling and introduces new strategies for aggregating responses from multiple proposers by a summarizer model. The key challenge is that if there are many proposers, a summarization model might not perform optimally due to context size limits and long context effects such as context rot or recency bias that shows up in long context scenarios. The authors propose methods to select a subset of proposers to overcome this, inspired by the combinatorial feature selection challenge in classical ML.

- The authors claim that since these classical feature selection methods are computationally infeasible in the LLM inference scenario, there is a need to develop efficient greedy algorithms that are more tractable in terms of compute requirements. The key insight leveraged to develop them is that proposers should be selected according to their joint compatibility with one another and with the summarizer, rather than relying on accuracy or diversity alone.

- In this paper they provide a counter-example showing that naive summarizer-agnostic critera can fail, and introduce three different greedy algorithms to address this limitation. Their experimental results show empirically that naive neuristics for aggregation can fail on certain benchmarks and the proposed complementarity oriented approaches often perform strongly and expose failure cases of simpler methods.

**Audience:**

Yes

**Audience Explanation:**

The paper documents some interesting observations which would be useful to researchers interested in ensembling LLM responses.

- Proposer usefulness depends on the summarizer. Figure 1’s phenomenon that the best standalone model may not be the best helper for a particular summarizer should be studied further given the large reported gaps.

- Simple selection heuristics fail in different regimes. Table 1 suggests that standalone accuracy, diversity, and LLM-judge scores can  be misleading. This is a useful negative result worth knowing.

**Claims And Evidence:**

No

**Claims Explanation:**

- Figure 1 only shows that the proposer producing the highest downstream accuracy can differ from the proposer with the highest standalone accuracy. The paper does not explain whether "most complementary" was selected on independent data - if it was selected by downstream accuracy on the same data, the comparison is partly circular.

- The results do not support "consistent robustness" across the proposed family. Table 1 shows truth-prediction and oracle-surrogate falling to 0.522 and 0.497 on reduced-pool AIME, well below input-all at 0.621.

- The claim that model-first frequently produces substantial gains over the baseline is a little weak as Table 1 shows limited gains.

- Model-first is allowed to search using the summarizer’s own validation accuracy, while the main baselines are not; without a budget-matched generic search baseline, the results show only that summarizer-aware search helps, not that the paper’s specific complementarity method is responsible.

- The paper states that it tested two summarizers per dataset but presents the better-performing summarizer in Table 1, so that table cannot by itself support claims covering all tested scenarios.

**Requested Changes:**

- Figure 1 selection protocol must be specified and evaluations should be out of sample. Choose the most complementary proposer on the validation set, then report its performance on a separate test set, alongside the most accurate proposer, for all summarizers and with uncertainty across repeated splits.

- Add a fair control for model-first. Compare it against ordinary forward greedy and random subset search using exactly the same number of summarizer calls. If model-first still wins, that supports the method rather than merely showing that access to summarizer feedback is useful.

- Replace Table 1 with the full result matrix and narrower claims. Report both summarizers for every dataset and pool in the main paper, then state exactly how often each method wins and by how much instead of using the better performing one only.